

# Comparative study of Cd uptake and tolerance of two Italian ryegrass (*Lolium multiflorum*) cultivars

Zhigang Fang[1,2], Laiqing Lou[1], Zhenglan Tai[1], Yufeng Wang[1], Lei Yang[1], Zhubing Hu[1] and Qingsheng Cai[1]

[1] College of Life Sciences, Nanjing Agricultural University, Nanjing, Jiangsu, China
[2] Kashgar University, Kashgar, Xinjiang, China

## ABSTRACT

Cadmium (Cd) is one of the most toxic heavy metals and is difficult to be removed from contaminated soil and water. Italian ryegrass (*Lolium multiflorum*), as an energy crop, exhibits a valuable potential to develop Cd polluted sites due to its use as a biofuel rather than as food and forage. Previously, via a screening for Cd-tolerant ryegrass, the two most extreme cultivars (IdyII and Harukaze) with high and low Cd tolerance during seed germination, respectively, were selected. However, the underlying mechanism for Cd tolerance was not well investigated. In this study, we comparatively investigated the growth, physiological responses, and Cd uptake and translocation of IdyII and Harukaze when the seedlings were exposed to a Cd (0–100 $\mu$M) solution for 12 days. As expected, excess Cd inhibited seedling growth and was accompanied by an accumulation of malondialdehyde (MDA) and reduced photosynthetic pigments in both cultivars. The effects of Cd on the uptake and translocation of other nutrient elements (Zn, Fe, Mn and Mg) were dependent on Cd concentrations, cultivars, plant tissues and elements. Compared with Harukaze, IdyII exhibited better performance with less MDA and higher pigment content. Furthermore, IdyII was less efficient in Cd uptake and translocation compared to Harukaze, which might be explained by the higher non-protein thiols content in its roots. Taken together, our data indicate that IdyII is more tolerant than Harukaze, which partially resulted from the differences in Cd uptake and translocation.

## INTRODUCTION

Heavy metal contamination in agricultural soil and water introduced by human activities poses a serious environment issue (*Bonfranceschi, Flocco & Donati, 2009*; *Mwamba et al., 2016*; *Toth et al., 2016*). Among heavy metals, cadmium (Cd), known as a highly toxic and non-biodegradable pollutant, is easily taken up by plant roots and translocated to other parts (*Nocito et al., 2011*), thus threatening human health via food contamination and ecosystem safety (*Agami & Mohamed, 2013*). In view of the difficulty of a cleanup of Cd-contaminated soil by physical or chemical means, planting energy crops became a viable alternative for exploiting heavy metal contaminated land (*Shi & Cai, 2009*;

Corresponding authors
Zhubing Hu,
zhubinghu@njau.edu.cn
Qingsheng Cai, qscai@njau.edu.cn

*Zhang et al., 2013*; *Al Chami et al., 2015*; *Pandey, Bajpai & Singh, 2016*). Thus, increasing efforts have been devoted to isolating a tolerant cultivar and dissecting the mechanisms underlying their tolerance.

Several direct and indirect toxic effects caused by excess Cd have been well-documented. First, excess Cd generates free radicals and reactive oxygen species (ROS), which can oxidize proteins, DNA, lipids and carbohydrates, thus disturbing a number of physical and biological processes in plants (*Belkhadi et al., 2010*; *Fernandez et al., 2013*). For instance, excess Cd induced an accumulation of a lipid peroxidation product, MDA, in rice, which is an indicator of oxidative stress and disturbed cellular metabolism (*Celekli, Kapi & Bozkurt, 2013*; *Mostofa, Seraj & Fujita, 2014*; *Xie et al., 2015*). Second, the uptake of nutrient elements (Fe, Mn, Cu and Zn) is disturbed by Cd stress, which can impair the transport of these elements from the roots to aerial parts, thus leading to a reduction of electron transport in photosystem I transport due to the lack of chlorophyll synthesis (*Aravind & Prasad, 2005*; *Lopes Júnior, Mazzafera & Arruda, 2014*).

A set of strategies in plants have been found to cope with exogenous Cd, which include active exclusion, vacuolar sequestration, retention in the roots, immobilization by cell walls and complexation by binding metal to low-molecular weight proteins (*Ramos et al., 2002*; *Wahid, Arshad & Farooq, 2010*). It has been reported that reducing Cd accumulation by exclusion in the roots of *Thlaspi arvense* conferred enhanced tolerance in the Cd-tolerance ecotype (*Martin et al., 2012*). In *Dittrichia viscosa* (L.) Greuter, the responses of Cd toxicity involved Cd retention in the cell wall of the roots and the upregulated contents of non-protein thiols (NPT) and organic acids (*Fernandez et al., 2014*). In wheat, Cd binds to the sulphur group of cysteine-forming Cd–phytochelatins complexes (Cd–PCs), reducing the free $Cd^{2+}$ in the cytosol, and the Cd–PCs complexes are in turn transported into the vacuole or out of the cell by ATP binding cassette transporters (*Greger et al., 2016*).

Different approaches have been employed to unravel the mechanisms that address Cd toxicity, such as screening for cadmium sensitive mutants (*McHugh & Spanier, 1994*) and dissecting the role of metal transporters by transgenic manipulations (*Uraguchi & Fujiwara, 2012*). The cultivar screen is another way to find evolved mechanisms in plants due to different environments and natural variations. This has been conducted for Cd tolerance and accumulation in several species such as hemp (*Shi et al., 2012*), Indian mustard (*Gill, Khan & Tuteja, 2011*), barley (*Sghayar et al., 2014*) and castor (*Zhang et al., 2014*), revealing that Cd tolerance is related to the characteristics of plant morphology, the amounts of phytochrome synthesis, Cd uptake and thiol levels.

Italian ryegrass (*Lolium multiflorum*), also called annual ryegrass, is broadly grown in the south of China during the winter before the emergence of rice to relieve green fodder shortages (*Ye et al., 2015*). Recently, this species has been considered an appropriate material for bio-ethanol production due to its high ethanol conversion, rapid growth and low input costs (*Yasuda et al., 2015*; *Ye et al., 2015*). Two recent studies reported that Italian ryegrass had a high tolerance to Cd during seed germination and was able to be cultivated in sites polluted by mine tailings (*Liu et al., 2013*; *Mugica-Alvarez et al., 2015*). In regards to these properties, Italian ryegrass has been suggested as a new species for the

bioremediation of heavy metal polluted soils (including Cd) (*Yamada et al., 2013*). However, compared with other species, little information is available concerning the capacity of Cd tolerance and uptake and tolerance mechanisms in Italian ryegrass. Here, through investigating the underlying causes for differential Cd tolerance in two ryegrass cultivars (IdyII and Harukaze), we found that IdyII is less efficient in Cd uptake and translocation than Harukaze. Furthermore, a high NPT content in IdyII might be one of causes for low Cd translocation. Our findings can provide a new tool for further dissecting the molecular mechanisms of Cd uptake and translocation in ryegrass cultivars and will be helpful for breeding Italian ryegrass as a bioenergy crop for heavy metal remediation.

## MATERIALS AND METHODS

### Plant cultivation

Two extreme ryegrass cultivars (IdyII and Harukaze) with high and low Cd tolerance during seed germination (*Fang et al., 2017*), respectively, were selected for this study. Seeds were sterilized with 10% $H_2O_2$ for 10 min, rinsed thoroughly with distilled water, and germinated via immersion in distilled water at 25 °C in the dark. After five days, uniform seedlings were transferred to 1 L plastic pots (14 plants per pot) filled with 1/4 Hoagland's solution. Seedlings were maintained for 10 days in a growth chamber at a 12 h light/dark cycle with 300 $\mu$mol m$^{-2}$ s$^{-1}$ light intensity, a day/night temperature of 25/20 °C and 65 ± 5% relative humidity. Five Cd concentrations for the treatments were as follows: 0, 5, 10, 25, 50, and 100 $\mu$M Cd as $CdCl_2 \cdot 2.5\ H_2O$ (analytical reagent) was added to the nutrient solution. Each treatment had six replicates. The nutrient solution was renewed every three days, and the pH was adjusted to 6.5 with 2 M NaOH or 2.7 M HCl.

After a 12-day Cd exposure, plants were divided into two groups and collected. One group was for biomass and Cd concentration determination, and the other group was for physiological index measurements, including chlorophyll content, lipid peroxidation and NPT. Each group had three replicates.

### Estimation of plant growth and Cd accumulation

The harvested plants were soaked in 20 mM $Na_2$-EDTA for 15 min, rinsed with distilled water to remove metals on the root surfaces, and separated into roots and shoots. Subsequently, samples were oven-dried at 70 °C overnight. The dried samples were weighed and digested with mixed acid [$HNO_3$ + $HClO_4$ (85:15, v/v)]. The concentrations of Cd, zinc (Zn), iron (Fe), manganese (Mn) and magnesium (Mg) were determined by an inductively coupled plasma optical emission spectrometer (ICP-OES, Optima 2100 DV; PerkinElmer, Inc., Waltham, MA, USA).

The tolerance indexes (TIs), translocation factors (TFs), bioconcentration factors (BCF), and Cd accumulation were determined according to the method of *Chen et al. (2011)*:

$$TIs = biomass_{Cd}/biomass_{control}$$

$$TFs = Cd_{concentration\ in\ shoot}/Cd_{concentration\ in\ root}$$

$$BCF = Cd_{concentration\ in\ root}/Cd_{concentration\ in\ the\ nutrient\ solution}$$

Cd accumulation = $[\text{biomass}]_{\text{dry weight}} \times [\text{Cd}]_{\text{concentration in plant tissues}}$

Total Cd accumulation = Cd accumulation in root + Cd accumulation in shoot

Cadmium distribution proportion of root = Cd accumulation in root/total Cd accumulation.

## Estimation of photosynthetic pigment contents

A total of 100 mg of the middle portion of fresh leaf slices were taken for pigments extraction with 10 mL 95% ethanol for 24 h in the dark. The absorbances of pigment extract were measured at wavelengths of 665, 649 and 470 nm with spectrophotometry (Shimadzu UV-2450; Shimadzu, Kyoto, Japan). Chlorophyll $a$ (Chl $a$), Chlorophyll $b$ (Chl $b$) and carotenoids (Car) were estimated using the following equations (*Knudson, Tibbitts & Edwards, 1977*), respectively:

Chl $a = 13.95 \times A_{665} - 6.88 \times A_{649}$

Chl $b = 24.96 \times A_{649} - 7.32 \times A_{665}$

Car $= (1{,}000 \times A_{470} - 2.05 \times \text{Chl } a - 114.8 \times \text{Chl } b)/245$

## Estimation of lipid peroxidation

The level of lipid peroxidation in root and leaf was estimated with 2-thiobarbituric acid (TBA) reactive metabolites chiefly malondialdehyde (MDA) as described by *Ali et al. (2014)*. Briefly, plant fresh tissues (0.1–0.3 g) were homogenized in 5 mL of 0.25% TBA made in 10% trichloroacetic acid. The extracts were heated at 95 °C for 30 min and then quickly cooled on ice. After centrifugation at 10,000$g$ for 10 min, the absorbance of the supernatant was measured at 532 and 600 nm. MDA contents was calculated with the following formula: MDA $= 6.45 \times (A_{532} - A_{600})$.

## Determination of NPT

Non-protein thiols were assayed following *Tian et al. (2011)* with minor modifications. Fresh tissues (approximately 0.3 g) were homogenized in 3 mL ice-cold 5% sulfosalicylic acid solution and centrifuged at 12,000$\times g$ (4 °C) for 15 min. The resulting supernatant was used for NPT assays. First, 0.3 mL of the supernatant was mixed with 1.2 mL 0.1 M K-phosphate buffer (pH 7.6) and 50 μL 6 mM 5,5'-dithiobis-2-nitrobenzoic acid (dissolved in 5 mM EDTA and 0.1 M phosphate buffer solution at pH 7.6). The mixture was incubated at room temperature for 20 min and then measured for absorbance at 412 nm with spectrophotometry. The NPT content was estimated with a standard curve of reduced glutathione in the range of 0–100 μg mL$^{-1}$.

## Statistical analysis

Statistical analyses were performed using a two-way analysis of variance (ANOVA) with SPSS version 20.0 (SPSS Inc., Chicago, IL, USA). Duncan's multiple range test was employed to compare the changes among the different treatments at $P < 0.05$. The correlations among TIs, Cd uptake, Cd TFs, Cd accumulation, MDA content and NPT

**Table 1 Effects of Cd on plant biomass, Cd tolerance, and the root/shoot ratio in two cultivars of Italian ryegrass.**

| Cultivar | Cd supply μM | Plant biomass (mg·plant$^{-1}$ DW) | | TIs | | Root/shoot ratio |
|---|---|---|---|---|---|---|
| | | R | S | R | S | |
| IdyII | 0 | 22.51 ± 2.77 a | 123.98 ± 10.40 a | | | 0.18 ± 0.012 a |
| | 5 | 22.60 ± 1.94 a | 120.19 ± 12.04 a | 1.00 ± 0.09 a | 0.97 ± 0.10 a | 0.19 ± 0.031 a |
| | 10 | 21.52 ± 2.33 ab | 113.92 ± 14.28 a | 0.96 ± 0.10 ab | 0.92 ± 0.12 ab | 0.19 ± 0.005 a |
| | 25 | 17.54 ± 1.09 bc | 103.17 ± 12.63 ab | 0.78 ± 0.05 bc | 0.83 ± 0.10 abc | 0.18 ± 0.036 a |
| | 50 | 13.30 ± 1.57 cde | 80.48 ± 8.41 bcd | 0.59 ± 0.07 d | 0.65 ± 0.07 c | 0.16 ± 0.003 ab |
| | 100 | 9.26 ± 0.99 ef | 78.08 ± 7.86 bcd | 0.41 ± 0.04 e | 0.63 ± 0.13 cd | 0.12 ± 0.014 b |
| Harukaze | 0 | 15.29 ± 0.99 cd | 84.00 ± 5.05 bc | | | 0.18 ± 0.023 a |
| | 5 | 14.64 ± 0.76 cd | 67.99 ± 2.92 cd | 0.96 ± 0.05 ab | 0.81 ± 0.03 bc | 0.22 ± 0.006 a |
| | 10 | 12.26 ± 0.90 de | 63.36 ± 3.64 cd | 0.80 ± 0.06 bc | 0.75 ± 0.04 bc | 0.19 ± 0.011 a |
| | 25 | 9.69 ± 0.70 ef | 55.06 ± 3.74 de | 0.63 ± 0.05 cd | 0.65 ± 0.04 c | 0.18 ± 0.010 a |
| | 50 | 5.46 ± 0.58 fg | 35.20 ± 3.62 ef | 0.36 ± 0.04 ef | 0.42 ± 0.04 de | 0.16 ± 0.017 ab |
| | 100 | 3.22 ± 0.13 g | 29.55 ± 1.43 f | 0.21 ± 0.01 f | 0.35 ± 0.02 e | 0.11 ± 0.007 b |
| ANOVA (F values) | Cd | 153.72** | 11.04** | 38.32** | 11.87** | 5.97* |
| | Cultivar | 25.79** | 92.04** | 15.15** | 18.93** | 0.017ns |
| | Cd × cultivar | 0.268ns | 0.13ns | 0.63ns | 0.24ns | 0.194ns |

Notes:
Values (mean ± SE, $n = 3$) with different letters in the same columns are significantly different according to the Duncan's multiple range test.
DW, dry weight; TI, tolerance index; R, root; S, shoot.
*$P < 0.05$, **$P < 0.01$, ***$P < 0.001$; ns, not significant.

content of the two cultivars in the roots in plant roots were determined by Pearson's correlation analysis.

# RESULTS

## Plant biomass, TIs and root/shoot ratio response to Cd stress

Increasing the Cd supply in the medium posed variable effects on plant biomass, TIs and the root/shoot ratio (Table 1). Cd treatments tended to reduce the biomass of both cultivars. The biomass of IdyII was significantly higher than that of Harukaze in the same treatment ($P < 0.01$), however biomass reductions with increasing Cd dose from 5 to 100 μM were not significant between IdyII and Harukaze. Similar alterations were also observed with the TIs. For example, when exposed to 25 μM Cd, root biomass was reduced by 37% in Harukaze and 22% in IdyII. A clear decline in the roots/shoot ratio was demonstrated in both cultivars with increasing Cd concentrations.

## Ecotoxicological response based on the plant biomass inhibition rate

As shown in Table 2, positive correlations were observed between the inhibition of plant biomass (root and shoot) and Cd concentrations in the solution ($P < 0.01$), which was represented by the quadratic equation. To evaluate toxicity, the inhibitory concentration (IC50; Cd concentration when the root or shoot biomass decreased by 50% compared with the control) and lethal concentration (IC90; Cd concentration when the root or shoot biomass decreased by 90% compared with the control) were determined by the fitting equation. The IC50 values of the shoots and roots of IdyII were 1.9-fold and
**Table 2 Fitted equations of Cd concentration and the inhibition rate of root or shoot biomass.**

| Cultivar | Fit equations between Cd concentration and inhibition rate of root biomass | IC$_{50}$ (μM) | IC$_{90}$ (μM) | $R^2$ | Fit equations between Cd concentration and inhibition rate of shoot biomass | IC$_{50}$ (μM) | IC$_{90}$ (μM) | $R^2$ |
|---|---|---|---|---|---|---|---|---|
| IdyII | $y = -0.0049x^2 + 1.1198x - 3.612$ | 68.27 | 192.03 | 0.991** | $y = -0.006x^2 + 0.9837x - 1.2897$ | 124.73 | 174.14 | 0.989** |
| Harukaze | $y = -0.01x^2 + 1.7983x - 0.8778$ | 35.17 | 121.58 | 0.994** | $y = -0.009x^2 + 1.4841x + 6.4125$ | 38.23 | 132.34 | 0.967** |

Notes:

$x$ is Cd concentration, and $y$ is inhabitation of root or shoot biomass. IC$_{50}$ indicates an effective Cd concentration (when the root or shoot biomass decreased by 50% compared with the control), and IC$_{90}$ indicates a lethal concentration (when the root or shoot biomass decreased by 90% compared with the control).
** Indicates $P < 0.01$.

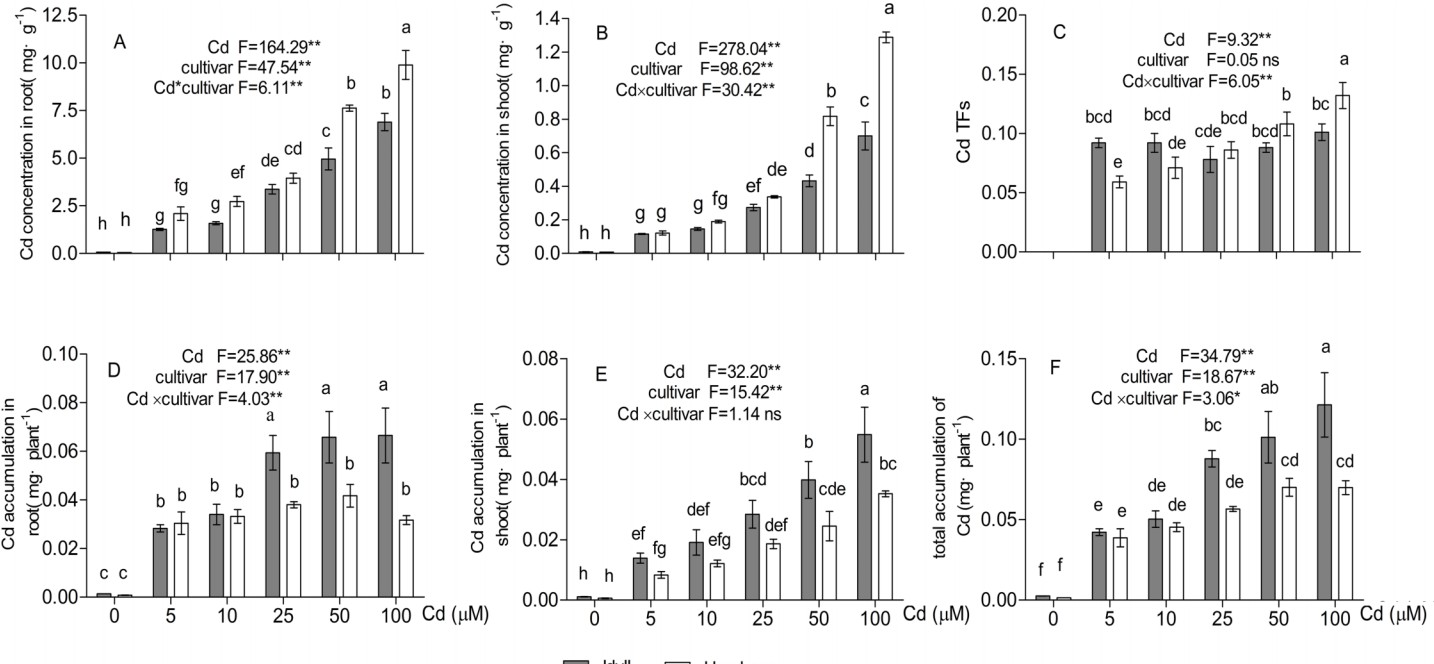

**Figure 1 Cadmium concentration, translocation factors (TFs) and cadmium accumulation in two cultivars of Italian ryegrass.** Data are mean ± standard error (SE) of three replicates. *$P < 0.05$, **$P < 0.01$. ns, not significant. Different letters indicate significant differences at $P < 0.05$ according to the Duncan's multiple range test.

3.2-fold higher than that of Harukaze, respectively. The lethal concentration of the roots and shoots (IC90) in IdyII were also higher than that of Harukaze, implying that IdyII was more tolerant to Cd compared to Harukaze.

## Cd concentration, accumulation in plant tissues, and BCFs and TFs responses to Cd stress

With elevating Cd concentrations in the treatment solutions, root Cd concentration in both cultivars increased, ranging from 2.09 to 9.89 mg·g$^{-1}$ in Harukaze and from 1.26 to 6.89 mg·g$^{-1}$ in IdyII. Cd concentrations in Harukaze roots were higher than that of IdyII roots, especially at the 50–100 μM Cd treatments (Fig. 1A, $P < 0.01$). Similar trends were also observed in the shoots (Fig. 1B). A gradual increase of Cd TFs in Harukaze was correlated with the increasing Cd concentration in the treatment solutions, whereas no significant Cd TFs changes in IdyII were observed. The Cd TF value in Harukaze was

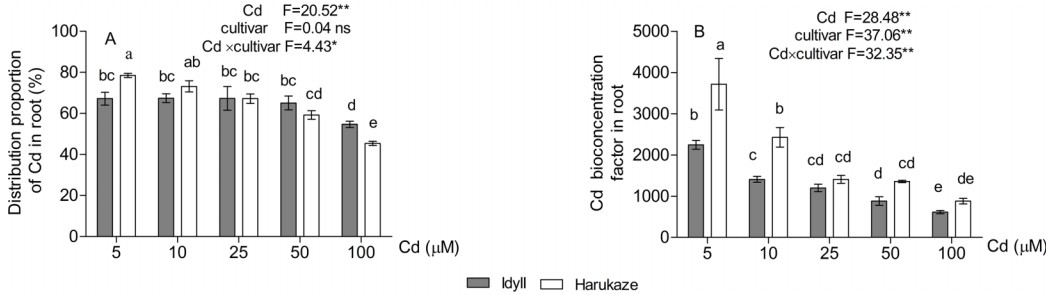

**Figure 2 The distribution proportion of Cd (A) and Cd bioconcentration factors (B) in roots under Cd stress.** Data are mean ± SE of three replicates. $^{*}P < 0.05$, $^{**}P < 0.01$. ns, not significant. Different letters indicate significant differences at $P < 0.05$ according to the Duncan's multiple range test.

significantly higher than that in IdyII after exposure to the highest Cd concentration (Fig. 1C, $P < 0.01$). Cd accumulation in the roots of Harukaze remained constant, while there was a dramatic increase in IdyII with the application of 25–100 μM Cd in the treatment solutions (Fig. 1D). An increasing trend of Cd accumulation in the shoots and total accumulation was observed with an increasing Cd supply, and their accumulation amounts in IdyII were significantly higher than that in Harukaze in the presence of higher Cd dosages (Figs. 1E and 1F, $P < 0.05$).

The proportion of cadmium distribution in the roots was 78.5–45.4% in Harukaze and 67.2–54.7% in IdyII under Cd stress. The distribution proportion in both Harukaze and IdyII significantly decreased with the 25 and 100 μM Cd treatments, respectively (Fig. 2A, $P < 0.01$). The increasing Cd supply reduced the root BCFs of the two cultivars (Fig. 2B), and the reduction was especially obvious in Harukaze (range from 3,715 to 880). At low Cd treatments (5–10 μM), the BCFs of Harukaze were markedly higher than that of IdyII ($P < 0.01$).

## Effects of Cd on plant mineral concentrations and TFs

Cadmium treatments altered the uptake and TFs of several nutrient elements (Fig. 3). Compared with the control, the 25–100 μM Cd supply markedly increased the Zn and Fe concentrations in the roots of Harukaze (Figs. 3A and 3B), whereas the Mn concentration in the roots of Harukaze was significantly decreased with lower Cd concentrations (5 and 10 μM) (Fig. 3C). Additionally, Mg concentration was greatly increased at the highest Cd concentrations (Fig. 3D). In contrast, compared with the control, 100 μM Cd significantly promoted Zn uptake in the roots of IdyII (Fig. 3A), while the uptakes of Fe, Mn, and Mg exhibited no change in the roots of IdyII (Figs. 3B–3D). In the shoot, Cd supply did not affect Zn and Mg concentrations in both cultivars, but severely decreased Mn concentrations (Figs. 3E, 3H and 3G). Compared with the control, shoot Fe concentrations in IdyII exhibited a gentle decrease with increasing Cd concentration, while no significant change occurred in Harukaze (Fig. 3F). In both cultivars, Zn TFs were significantly inhibited at the 25–100 μM Cd treatments (Fig. 3I), and the amounts in IdyII were significantly higher than that of Harukaze. Fe TFs reached a maximum in both cultivars under 5 μM Cd and then showed a decrease with increasing Cd (Fig. 3J).

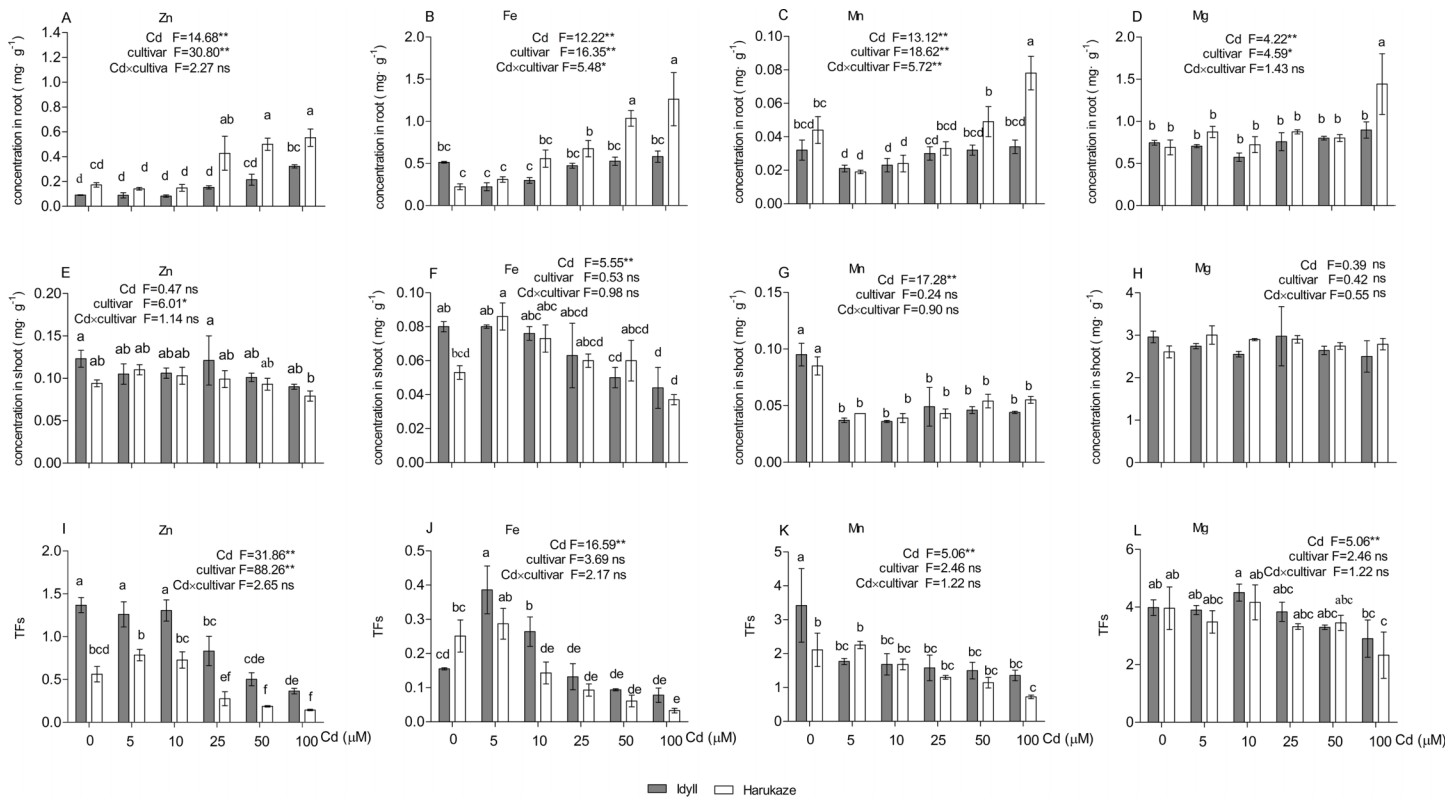

**Figure 3 Nutrient element concentrations in the roots, shoot and TFs in the presence of Cd.** Data are mean ± SE of three replicates. $^{*}P < 0.05$, $^{**}P < 0.01$. ns, not significant. Different letters indicate significant differences at $P < 0.05$ according to the Duncan's multiple range test.

Cd treatments in IdyII significantly reduced Mn TFs and had no change in Mg TFs. In contrast, the TFs of Mn and Mg in Harukaze decreased considerably only at the highest concentration of Cd (Figs. 3K and 3L). According to a two-way ANOVA analysis, significant differences in the concentrations of Fe and Mn ($P < 0.01$) of the roots were found between the two cultivars, while no significant interaction were observed for other parameters (Figs. 3A–3L), indicating these two cultivars only respond significantly different for the uptakes of Fe and Mn when exposed in excess Cd.

## Effects of Cd on pigment content, lipid peroxidation and NPT content

The cadmium supply tended to reduce Chl $a$, Chl $b$, Chl ($a + b$) and Car contents in both cultivars (Table 3). For example, the Chl $a$, Chl $b$, and Car content decreased by 42.5%, 44.7% and 44.4% in Harukaze and by 11.8%, 5.9%, and 22.2% in IdyII under 25 μM Cd stress, respectively. The Chl $a$, Chl $b$ and Chl ($a + b$) content of IdyII was significantly higher than that of Harukaze at 25–100 μM Cd concentrations ($P < 0.01$); a similar trend occurred in Car under 50–100 μM Cd stress. The Cd treatments did not affect the Chl $a/b$ ratio of IdyII but significantly reduced the Chl $a/b$ ratio of Harukaze at 100 μM Cd.

The MDA content in plant tissues was increased with elevated Cd concentrations, and the amounts in the leaves were higher than in the roots (Fig. 4A, $P < 0.01$). Compared with the control, when treated with high Cd concentrations (50 and 100 μM), the MDA content of the seedling roots increased by 252.5% and 610.2% in Harukaze but only by

**Table 3 Effects of Cd on the photosynthetic pigments in the leaves of two Italian ryegrass cultivars.**

| Cultivar | Cd supply μM | Chl $a$ mg·g$^{-1}$ FW | Chl $b$ mg·g$^{-1}$ FW | Car mg·g$^{-1}$ FW | Chl $(a + b)$ mg·g$^{-1}$ FW | Chl $a/b$ mg·g$^{-1}$ FW |
|---|---|---|---|---|---|---|
| IdyII | 0 | 1.10 ± 0.07 abc | 0.34 ± 0.01 ab | 0.18 ± 0.01 abc | 1.44 ± 0.05 ab | 3.25 ± 0.02 ab |
| | 5 | 1.03 ± 0.09 bcd | 0.32 ± 0.02 abc | 0.16 ± 0.02 abcd | 1.36 ± 0.07 bc | 3.21 ± 0.05 abc |
| | 10 | 0.97 ± 0.22 cd | 0.31 ± 0.04 bc | 0.16 ± 0.02 bcde | 1.28 ± 0.16 bc | 3.08 ± 0.08 abcd |
| | 25 | 0.97 ± 0.05 cd | 0.32 ± 0.02 abc | 0.14 ± 0.03 cde | 1.33 ± 0.04 bc | 3.05 ± 0.22 abcd |
| | 50 | 0.86 ± 0.01 d | 0.27 ± 0.01 c | 0.13 ± 0.01 cde | 1.13 ± 0.02 c | 3.13 ± 0.12 abc |
| | 100 | 0.58 ± 0.03 ef | 0.20 ± 0.01 d | 0.10 ± 0.01 ef | 0.79 ± 0.01 de | 2.86 ± 0.20 bcd |
| Harukaze | 0 | 1.20 ± 0.12 ab | 0.38 ± 0.02 a | 0.19 ± 0.01 ab | 1.60 ± 0.10 a | 3.17 ± 0.08 abc |
| | 5 | 1.22 ± 0.06 a | 0.38 ± 0.01 a | 0.20 ± 001 a | 1.60 ± 0.05 a | 3.18 ± 0.04 abc |
| | 10 | 0.95 ± 0.12 cd | 0.29 ± 0.02 bc | 0.14 ± 0.01 cde | 1.24 ± 0.09 bc | 3.34 ± 0.04 a |
| | 25 | 0.69 ± 0.06 e | 0.21 ± 0.01 d | 0.12 ± 0.01 def | 0.89 ± 0.04 d | 3.30 ± 0.03 a |
| | 50 | 0.48 ± 0.11 f | 0.17 ± 0.02 d | 0.08 ± 0.01 f | 0.65 ± 0.08 e | 2.83 ± 0.03 cd |
| | 100 | 0.21 ± 0.08 g | 0.07 ± 0.02 e | 0.03 ± 0.01 g | 0.29 ± 0.06 f | 2.71 ± 0.21 d |
| ANOVA (*F* Values) | Cd | 51.73*** | 42.84*** | 20.56*** | 51.17*** | 4.43** |
| | Cultivar | 15.04* | 19.39*** | 4.92* | 16.73*** | 0.01ns |
| | Cd × cultivar | 9.584*** | 10.24*** | 5.39** | 10.18*** | 1.86ns |

**Notes:**
Chl $a$, Chl $b$, and Car, indicate Chl $a$, Chl $b$, and carotenoids, respectively. Values (mean ± SE; $n = 3$) with different letters in the same columns are significantly different according to the Duncan's multiple range test.
*$P < 0.05$, **$P < 0.01$, ***$P < 0.001$; ns, not significant.

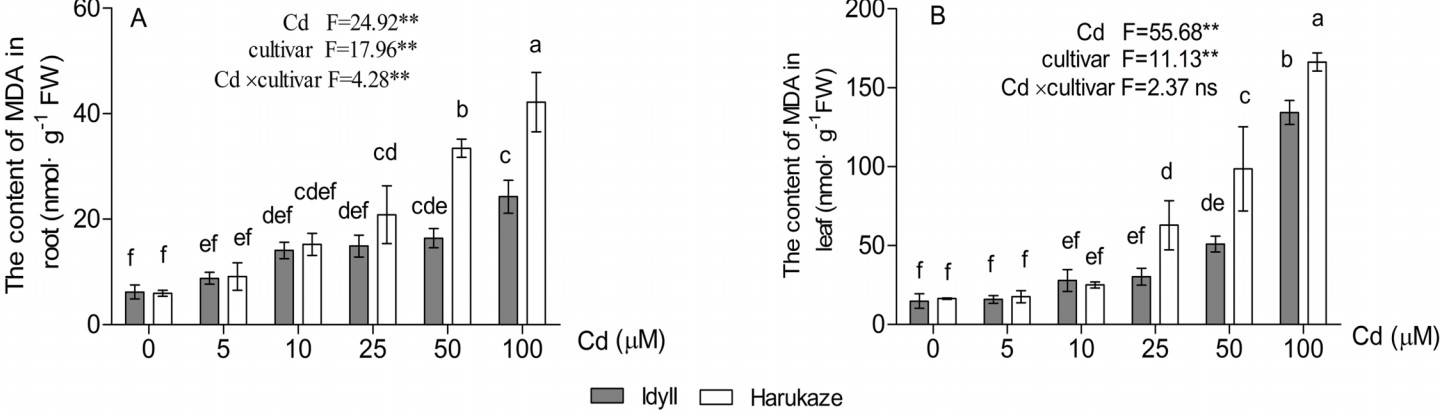

**Figure 4 Effects of Cd on malondialdehyde (MDA) content in the roots and shoots of two Italian ryegrass cultivars.** Data are mean ± SE of three replicates. *$P < 0.05$, **$P < 0.01$. ns, not significant. Different letters indicate significant differences at $P < 0.05$ according to the Duncan's multiple range test.

140.7% and 291.7% in IdyII, respectively. Similarly, the MDA content of the leaves in Harukaze increased sharply with the 25–100 µM Cd treatments and was considerably higher than that of IdyII (Fig. 4B, $P < 0.01$).

As shown in Fig. 5, compared with the control, the NPT content in the roots of IdyII increased at least three times with the application of 5 µM Cd and reached its maximum under 50 µM Cd. No significant change occurred in Harukaze with the addition of 5–50 µM Cd in the medium (Fig. 5A). Furthermore, the NPT content in the roots of IdyII were significantly higher than that of Harukaze with the treatments of 10–100 µM

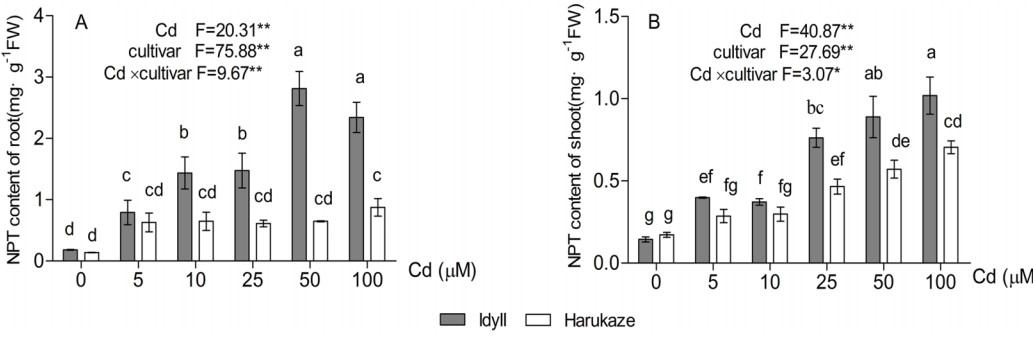

**Figure 5 Effects of Cd on non-protein thiols (NPT) content in the roots and shoots of two Italian ryegrass cultivars.** Data are mean ± SE of three replicates. *$P < 0.05$, **$P < 0.01$. ns, not significant. Different letters indicate significant differences at $P < 0.05$ according to the Duncan's multiple range test.

**Table 4 Correlation coefficients ($n = 30$) among Cd TIs, uptake, translation, MDA, NPT in Italian ryegrass roots.**

| Index | TIs | Cd concentration | TFs | Cd accumulation | MDA content | NPT content |
|---|---|---|---|---|---|---|
| Tis | 1 | | | | | |
| Cd concentration | −0.922** | 1 | | | | |
| TFs | −0.612* | 0.635* | 1 | | | |
| Cd accumulation | −0.185 | 0.293 | −0.003 | 1 | | |
| MDA content | −0.796** | 0.837** | 0.722** | 0.054 | 1 | |
| NPT content | −0.143 | 0.147 | 0.04 | 0.738** | −0.072 | 1 |

Notes:
TIs, TFs, MDA, and NPT indicate tolerance indexes, translocation factors, malondialdehyde, and non-protein thiols, respectively.
*$P < 0.05$, **$P < 0.01$.

Cd ($P < 0.01$). The NPT content in the shoots of the two cultivars were also enhanced under Cd stress. The values in IdyII were significantly higher than that in Harukaze at the 25–100 µM Cd treatments (Fig. 5B, $P < 0.05$). By a two-way ANOVA analysis, a significant interaction was found (Figs. 5A and 5B), demonstrating that both cultivars responded significantly different to Cd treatments for NTP production in root and shoot.

### Correlation analysis

Pearson's correlation analysis was carried out to investigate the correlations among TIs, Cd uptake, Cd TFs, Cd accumulation, MDA content and NPT content of the two cultivars in the roots (Table 4). TIs were negatively correlated to Cd concentration, MDA content and Cd TFs. Cd uptake and TFs were positively correlated to MDA content. Additionally, a positive correlation was observed between Cd accumulation and NPT content.

### DISCUSSION

In this study, our data demonstrated that Italian ryegrass possesses a stronger capacity in Cd uptake than common crops. After 12 days of exposure at 50 µM Cd, the Cd concentration in Italian ryegrass reached at least 4.9 mg·g$^{-1}$ (DW) in the roots and 0.4 mg·g$^{-1}$ (DW) in the shoots (Figs. 1A and 1B). These concentrations are higher

than that in rice after 15 days of exposure at 50 µM Cd (*Lin et al., 2012*), which is 0.35 mg·g$^{-1}$ (DW) in the roots and 0.15 mg·g$^{-1}$ (DW) in the shoots, as well as that in maize after 15 days of exposure at 100 µM Cd (*Wang et al., 2007*), which is 1.95 mg·g$^{-1}$ (DW) in the roots and 0.35 mg·g$^{-1}$ (DW) in the shoots.

The biomass reduction in the roots was more visible than that in the shoots when Cd levels were over 10 µM (Table 1). Similar results were reported in barley (*Tiryakioglu et al., 2006*), and the reason may be the fact that the roots are directly exposed to Cd (*Hegedüs, Erdei & Horváth, 2001*). As plant biomass and TIs are two important parameters to evaluate the Cd tolerance in plants (*Metwally et al., 2005*; *Shi et al., 2012*), the biomass of the roots and shoot in Harukaze and TIs were reduced over 50% under 50 and 100 µM Cd, which was not observed in IdyII (Table 1), thus demonstrating that IdyII was more tolerant to Cd than Harukaze. This was further supported by the higher IC50 and IC90 of Cd toxicity in IdyII (Table 2), which are two parameters commonly representing phytotoxin under a threshold and acute toxicity, respectively (*Paschke, Valdecantos & Redente, 2005*; *An, 2006*; *Pannacci, Pettorossi & Tei, 2013*).

Although Cd is a non-redox metal unable to produce ROS through single election transfer, Cd could interfere with the antioxidant defence system by replacement of metal cofactors enzymes, leading to the diminishment of the capacity for ROS removal (*Wahid, Arshad & Farooq, 2010*). Cd also affects the functions of two important organelles, the mitochondria and chloroplasts, which in turn disturb their election transfers and generate free radicals and ROS in the cell (*Celekli, Kapi & Bozkurt, 2013*; *Mostofa, Seraj & Fujita, 2014*). The accumulated ROS can interact with proteins, lipids, carbohydrates, and DNA, perturbing a number of physiological processes (*Gallego et al., 2012*). In IdyII and Harukaze, the Cd supply enhanced the MDA content, indicating Cd induced the oxidation of lipids (Fig. 4). The lipid peroxidation might partially be attributed to the reduction in photosynthetic pigments under Cd stress in both cultivars (Table 3). Relatively high MDA content and low photosynthetic pigments demonstrated that Cd-induced toxicity in Harukaze was more severe than in IdyII, which was consistent with their Cd tolerance. Similar correlations between Cd tolerance and MDA content were observed in oilseed cultivars (*Wu et al., 2015*), the leaves of Indian mustard cultivars (*Gill, Khan & Tuteja, 2011*) and Artichoke cultivars (*Chen et al., 2011*).

Apart from oxidative damage, the uptake, transport, and subsequent distribution of nutrient elements in IdyII and Harukaze were affected by the presence of Cd (Fig. 3). An elevated Cd dosage increased Zn, Fe, Mg and Mn concentrations in Harukaze roots, whereas it did not significantly increase that in the IdyII roots, except for Zn (Figs. 3A–3D). Possibly, the metal transportation systems in the roots are different between Harukaze and IdyII, and Harukaze may have a high-dose Cd activated transportation system. This is further supported by the higher Cd concentrations in Harukaze. Likewise, Cd promoting the uptake of Mg, Ca, and Fe were reported in tomato (*Kisa, Ozturk & Tekin, 2016*). No significant differences in metal concentration were observed between the shoots of Harukaze and IdyII, indicating that cultivar differences in metal uptake are mainly in the roots rather than in the shoots. *Goncalves et al. (2009)* also suggested that microelement uptake, such as Fe$^{2+}$, Mn$^{2+}$ and Zn$^{2+}$, was determined by

the level of Cd in the substrate, cultivar and plant tissue specificity in potato (*Solanum tuberosum*). Several metal transporters have been identified that translocate nutrient elements from the roots to the shoot, such as NRAMP families and ZIP families (*Choppala et al., 2014*). With exposure to Cd, the Cd TF remained constant in IdyII and increased in Harukaze along with increasing concentration (Fig. 1C), whereas the TFs of Zn, Fe and Mn exhibited a decline (Figs. 3I–3K), indicating that there may be possible competition with the metal transporters for translocation between Cd and other micronutrients in Italian ryegrass. It was reported that there were antagonistic effects from Cd and microelement elements (Zn, Fe, Mn) using the same transporters and/or cation channels as Ca and Mg (*Sarwar et al., 2010*; *Kisa, Ozturk & Tekin, 2016*).

Non-protein thiols, including glutathione, thiol-rich peptides and other SH groups, play an important role in defence response against the detoxification of heavy metals in plants (*Ozdener & Aydin, 2009*; *Nadgorska-Socha et al., 2013*). In our study, Cd concentration in the roots and shoots of Harukaze were significantly higher than that of IdyII, whereas the NPT content was lower in Harukaze than in IdyII (Figs. 1A, 1B and 5). NPT are essential for the synthesis of Cd-binding peptides such as phytochelatins, which inactivate and sequester Cd by forming stable Cd-complexes in the vacuole (*Cobbett, 2000*). The high NPT in IdyII may promote Cd sequestration into the vacuole and block its translocation, thus leading to the decline of the Cd TF. A similar phenomenon was observed in the variation of Cd tolerance among cultivars of cabbage and barley, suggesting that NPT content may be an important indicator for Cd tolerance (*Tiryakioglu et al., 2006*; *Sun et al., 2013*). Except for Cd, the higher NPT content in IdyII might be also contributed to the decreased translocation of Zn from the roots to the shoots (Fig. 3I). Furthermore, in comparison with Harukaze, a lower Cd concentration in the root of Idyll indicates that Idyll also is an excluder that does not take up and transport Cd easily, the underlying mechanism of which will need to be further investigated.

## CONCLUSION

In the present study, the biomass, Cd uptake, translocation, accumulation, and physiology parameters of two Italian ryegrass cultivars were significantly affected by Cd treatments. Compared with Harukaze, IdyII is a Cd-tolerant cultivar, exhibiting a low Cd uptake and a high NPT content. These two distinct capacities may be the major physiological changes that contributed to the difference of Cd tolerance between the two cultivars. Taken together, our data demonstrates that IdyII is more tolerant than Harukaze, which is correlated with low Cd uptake and high NPT content. This will be helpful in investigating the molecular mechanisms of Cd uptake and translocation in Italian ryegrass.

## ACKNOWLEDGEMENTS

We would like to thank Chunhua Zhang, who is working at the Experimental Center of Life Science, Nanjing agricultural university, for samples determination during phases of the project. We thank the seeds provided by Chenglong Ding, manager of Livestock Science, Jiangsu Academy of Agricultural Sciences. We also thank Dr. Gaoling Shi for his help with discussion and comments on the early version of this manuscript.

### Funding

This work was supported by grants from the Jiangsu Science and Technology Support Program for Social Development (No. BE2014709) and the Natural Science Foundation of China (NSFC, 31372359). The funders had no role in study design, data collection and analysis, decision to publish, or preparation of the manuscript.

### Grant Disclosures

The following grant information was disclosed by the authors:
Jiangsu Science and Technology Support Program for Social Development: BE2014709.
Natural Science Foundation of China: 31372359.

### Competing Interests

The authors declare that they have no competing interests.

### Author Contributions

- Zhigang Fang conceived and designed the experiments, performed the experiments, analyzed the data, contributed reagents/materials/analysis tools, wrote the paper, prepared figures and/or tables.
- Laiqing Lou analyzed the data, contributed reagents/materials/analysis tools.
- Zhenglan Tai performed the experiments.
- Yufeng Wang performed the experiments.
- Lei Yang performed the experiments.
- Zhubing Hu analyzed the data, wrote the paper, prepared figures and/or tables, reviewed drafts of the paper.
- Qingsheng Cai conceived and designed the experiments, wrote the paper, reviewed drafts of the paper.

### Data Availability

    The raw data has been supplied as Supplemental Dataset Files.

### Supplemental Information

Supplemental information for this article can be found online at http://dx.doi.org/10.7717/peerj.3621#supplemental-information.

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
