# Peer review of "Comparative study of Cd uptake and tolerance of two Italian ryegrass (Lolium multiflorum) cultivars"

_PeerJ, doi:10.7717/peerj.3621_

## Round 0.1 · original submission · Major Revisions

It is clear from both reviews that the current manuscript is not suitable for publication. Several problems need to be remedied.

- The English is substandard and needs to be improved, by having it edited by a native English speaker. In case you don't have someone available there are commercial services to provide this service.
- The introduction needs to better state the current knowledge of the Cd response of Italian ryegrass and/or related species.
- The statistics needs to be described better in the materials and methods and included in all figures.
- The extrapolation from the results obtained here in hydroponics and application towards phytoremediation needs to be better supported (e.g. by adding soil/pot or field experiments) or toned down.

Reviewer 1 ·

Basic reporting

1. Basic Reporting
• The submission must adhere to all PeerJ policies (see: 'Journal Policies').
• The article must be written in English using clear and unambiguous text and must conform to professional standards of courtesy and expression.

The level of English used in this article is not good enough to be published. It should definitely be reviewed and rewritten by a native speaker with a scientific background. It cannot be published in this condition.



• The article should include sufficient introduction and background to demonstrate how the work fits into the broader field of knowledge. Relevant prior literature should be appropriately referenced.

Short introduction: the introduction could be improved by providing more relevant background.

Also, specify the physiological stress markers you are going to address in your article.



• The structure of the submitted article should conform to an acceptable format of ‘standard sections’ (see our Instructions for Authors for our suggested format). Significant departures in structure should be made only if they significantly improve clarity or conform to a discipline-specific custom.

Standard sections: OK

The results of the effect of Cd on plant mineral concentrations and TFs are badly structured and it is hard for the reader to comprehend. My advice is to split up the results for root, shoot and TF and discuss the cultivars separately for each of the minerals. It is not needed to introduce new titles, but you can work with paragraphs here.



• Figures should be relevant to the content of the article, of sufficient resolution, and appropriately described and labeled.

The labels of the figures are incomplete. A part of the text is missing at the end.

Figure 3 also does not mention anything about the p-values or levels of significance, nor does it mention the nutrients (metals) shown in the graphs.

There is also no information about the error bars in each of the figures. Are they STDEV or SE or … ?

In some figures, there is only “g” as unit (for example in figure 3), where it should be “mg / g FW”).

The image quality of the graphs is OK.



• The submission should be ‘self-contained,’ should represent an appropriate ‘unit of publication’, and should include all results relevant to the hypothesis. Coherent bodies of work should not be inappropriately subdivided merely to increase publication count.

In my opinion, the submission is self-contained. The paper informs me on the tolerance of two Italian ryegrass cultivars. It also addresses one mechanism to cope with the Cd stress.

The article has no revolutionary conclusions (in my opinion).

Experimental design

2. Experimental Design

• The submission must describe original primary research within the Aims & Scope of the Journal.

To my knowledge, it is original primary research.



• The submission should clearly define the research question, which must be relevant and meaningful. The knowledge gap being investigated should be identified, and statements should be made as to how the study contributes to filling that gap.

No stated knowledge about the effect of Cd on Italian ryegrass in other literature. This article gives information about the effect of Cd on Italian ryegrass.



• The investigation must have been conducted rigorously and to a high technical standard.

The number of replicates for each treatment is quite small. Still, the error bars are small. BUT: The authors do not indicate which calculation they have used for the error bars.



• Methods should be described with sufficient information to be reproducible by another investigator.

Information about the calculation of distribution proportion is not given.

Information about the correlation test is not given in M&M.

The formulas of the TFs and BCFs are incomplete. It is not stated whether Cd concentration, Cd accumulation, … is used for these calculations.



• The research must have been conducted in conformity with the prevailing ethical standards in the field.

OK.

Validity of the findings

3. Validity of the Findings

• The data should be robust, statistically sound, and controlled.

Only 3 biological replicates for each treatment. Yet, they have achieved to obtain small error bars. BUT: The others do not indicate which calculation they have used for the error bars.



• The data on which the conclusions are based must be provided or made available in an acceptable discipline-specific repository.

Data are supplied.

I did have insufficient time to check whether these data and calculations are correct.



• The conclusions should be appropriately stated, should be connected to the original question investigated, and should be limited to those supported by the results.

Conclusion is in line with the research question.



• Speculation is welcomed, but should be identified as such.

/



• Decisions are not made based on any subjective determination of impact, degree of advance, novelty, being of interest to only a niche audience, etc.

Reconsider the use of Italian ryegrass for cultivation and phytoremediation.

Looking at fundamental knowledge about the Cd tolerance mechanisms is interesting.

If you want to look at these mechanisms for cultivation of Italian ryegrass for fodder, fuel, ... you should first consider the fact that it is grown on Cd contaminated soil.

The use of Italian ryegrass for phytoremediation should be reconsidered since accumulation levels in root and shoot are about the same.

For example, in contrast to maize, see: Wang, Aiyun, Minyan Wang, Qi Liao, and Xiquan He. 2016. “Characterization of Cd Translocation and Accumulation in 19 Maize Cultivars Grown on Cd-Contaminated Soil: Implication of Maize Cultivar Selection for Minimal Risk to Human Health and for Phytoremediation.” Environmental Science and Pollution Research 23 (6): 5410–19. doi:10.1007/s11356-015-5781-z.

In this article, they have shown that maize accumulates about 80% in the straw and grain.



• Replication experiments are encouraged (provided the rationale for the replication, and how it adds value to the literature, is clearly described); however, we do not allow the ‘pointless’ repetition of well known, widely accepted results.

No replication, since the authors state that it is not yet tested for their cultivars / Italian ryegrass.



• Negative / inconclusive results are acceptable.

I would reconsider being positive about the cultivation purposes for Italian ryegrass (phytoremediation, fodder, …). You might want to add a negative advice for that. You could consider fundamental research concerning the mechanisms in tolerance, damage, uptake and transportation.

Additional comments

I have added the PDF with markings and annotations.

The red markings concern syntax errors. I have not marked all the syntax errors.

The orange markings are questions or remarks. The questions/remarks are added to these markings.

The large blue marking (of Effects of Cd on plant mineral concentrations and TFs) concerns a block of text which is hard to comprehend. I advise a new structure for that text in order to make it easier to comprehend.

If you cannot read these remarks with your pdf-viewer, please open the pdf with "Adobe Acrobat Reader" and open the remark tab. Please do read the remarks.


Also: In my opinion, Italian ryegrass is not suitable for phytoremediation or cultivation for fodder or fuel (on Cd contaminated soil) (based on your results about accumulation). Since I'm quite new in this field of research, I do advise you to seek out more peers and discuss this matter with them.



I do think your research is worth being published, since you compare a tolerant and a sensitive cultivar and their response to Cd. Maybe address their variation in response in the conclusion and be specific about it (mention the parameters you've measured).


I hope you find my review valuable and I also hope you are able to use this feedback in improving your manuscript.

Annotated reviews are not available for download in order to protect the identity of reviewers who chose to remain anonymous.

Reviewer 2 ·

Basic reporting

The manuscript describes the effect of Cd exposure in 2 Italian ryegrass cultivars with regard to Cd uptake and translocation as well as the sensitivity/tolerance characteristics of both cultivars. The authors performed an extensive study with plenty of data but at present there are some major remarks that should be addressed by the authors.

Concerning the English language, the manuscript should be carefully checked for grammatical errors, such as use of tenses, plural/singular... It is recommended to have the manuscript checked by a native English speaking person to improve the readability of the manuscript.

Experimental design

Whereas the authors want to investigate the underlying mechanisms of Cd responses in 2 different ryegrass cultivars in relation to biomass production and Cd uptake for phytoremediation purposes, a hydroponic setup might be questioned.
The authors should be careful with stating that high Cd accumulation occurs in hydroponics, this might definitely not be the case when plants are grown on Cd-contaminated soils. Extrapolation of lab results (controlled environmental setup) to field experiments is not straightforward. It is highly recommended to add an experiment on these cultivars grown on Cd-contaminated soils to confirm or not the possible extrapolation of the current data.

In the majority of the M&M section, the authors describe very well the methodology, only for Cd distribution proportion, a description is lacking. This should be added by the authors.

Validity of the findings

Although the authors performed a lot of work, the novelty of the manuscript is rather restricted as the analyses performed under different Cd exposures have been explored in multiple studies. Nevertheless, the work published on Lolium multiflorum is limited.

Although the figures are clear, the authors should add the statistical analysis in the figure legends (as was done for figure 1).

With respect to phytoremediation purposes, the current manuscript should be considered with care because hydroponics and metal uptake are completely different from field conditions on Cd-contaminated soils. The authors should revise this way of description of the context for there research questions.

---

## Round 0.2 · Minor Revisions

The response letter is not detailed enough. In most places you just indicate that specific parts have been rewritten. You need to specify the change you made and the line numbers where this is done.

We requested the English to be corrected by a NATIVE English speaker or service, it is apparent that you got the help from a colleague that spent time abroad. Although this is an improvement, I still encountered numerous grammatical errors and unclear sentences, so the manuscript still needs further improvement before I'm willing to resend it for review.

---

## Round 0.3 · Minor Revisions

Both reviewers are satisfied by the way you addressed their main concerns. One has a comprehensive list of detailed remarks that I suggest you address as they could help to remove errors and increase clarity.

Reviewer 1 ·

Basic reporting

• English language: OK, but need minor changes here and there (be sure to look at my specific remarks for the individual lines). The editorial editing from AJE greatly improved the readability.


• Literature references: OK, sufficient.


• Raw data:

o raw_data1: translate foreign language + Re in first column is unclear + file is coupled to other file, please make file work on its own

o raw_data2: OK

o raw_data3: translate foreign language

o raw_data4: name sheet 1

o raw_data5: name sheet 1

o raw_data6: translate foreign language

o I was unable to find absorbance levels for NPT at 412 nm. I might have overlooked it.


• Figures and tables:

o Figure 1:
Isn’t it: “Duncan’s multiple range test”?
“(n = 3) of three replicates”  don’t they mean the same?

o Figure 2:
Isn’t it: “Duncan’s multiple range test”?
“(n = 3) of three replicates”  don’t they mean the same?

o Figure 3:
Isn’t it: “Duncan’s multiple range test”?
“(n = 3) of three replicates”  don’t they mean the same?
It would might help to add Zn, Fe, Mn and Mg above the graphs and Root concentration, Shoot concentration and TFs vertically to the left of the graphs. Just so the reader gets a quick overview of the contents of all the graphs.

o Figure 4:
Isn’t it: “Duncan’s multiple range test”?
“(n = 3) of three replicates”  don’t they mean the same?

o Figure 5:
Isn’t it: “Duncan’s multiple range test”?
“(n = 3) of three replicates”  don’t they mean the same?

o Table 1:
DW, dry weight
Declare what the values mean (i.e. 153.73, 11.04, 38.32, …)
Isn’t it: “Duncan’s multiple range test”?

o Table 2:
inhabitation?

o Table 3:
Duncan…

o Table 4:
OK

Experimental design

• Identified knowledge gap: OK

• Research question is relevant and meaningful.

• High technical and ethical standard: OK

• Methods: be sure to look at my specific remarks for the individual lines.

Validity of the findings

OK + be sure to look at my specific remarks for the individual lines.

Additional comments

Dear authors,

Your current article has improved a lot compared to the previous version. The text was clear and the level of English was very good (except for a few sentences).
Please consider the changes and adaptations which I suggested, but also feel free to argument why something should not be changed or adapted, since it is just my opinion.
The approach to a more fundamental report of the response of the two cultivars to Cd was an improvement and most of the discussion is well written.
I have chosen for a major revision, even though the revision needed is of a much smaller scale compared to the previous version. Since the article has changed a lot compared to the previous version, I would like to go through it once more after you have dealt with the remarks of the reviewers. By opting for a major revision, I get a chance to see how you interpreted my comments for improvement and where you do not agree (which of course is possible as well ! ).
Good luck and thank you for your effort.

Kind regards.


IMPORTANT:
I have specific remarks for individual lines. These remarks concern basic reporting, experimental design & methods and validity of the findings & discussion.

These remarks are:
Line 65: It is not just because a substance is toxic to plants, that plants have evolved mechanisms to cope with it.
Line 73: Cd-PC, this abbreviation is stated or explained elsewhere.
Line 121: “to the constant weight” --> the?
Line 134: “the middle part of 100 mg of fresh leaves” --> what do you mean?
Line 135: Link wave lengths to what you measure at these wave lengths
Line 139: What is extracted?
Line 142: What do you measure at these wave lengths?
Line 142: Don’t you just measure MDA as a parameter for lipid peroxidation. Here it seems like you have measured both.
Line 158: “some physiology parameters” --> to vague
Line 165 – 166: “biomass reductions with increasing Cd dose from 5 μM to 100 μM were more distinct in Harukaze (P<0.01). --> ANOVA Cdxcultivar is not significant, meaning that both cultivars do not react differently to the cadmium treatment. Cultivars are significantly different, but they already differ in starting weight, which might be the most important factor why the weight of the cultivars differs.
Line 173: Is it EC or IC50? I’m not sure. Please check.
Line 178-179: Please nuance! Idyll is not tolerant, but MORE tolerant to Cd compared to Harukaze.
Line 183: Why is it obvious? Sounds like you were expecting this or sounds like there is a specific reason for this.
Line 185: Correlation is significant? Table 4 shows correlation both cultivars taken together. Could it be interesting to check correlations for both cultivars separately?
Line 187: TFs --> TF
Line 211: Reduction is not significant
Line 217 – 219: More importantly: The 2 cultivars only respond significantly different for Fe and Mn in the root. There, the interaction of Cd and cultivar is significant. For the other parameters, there is no significant interaction, meaning there is no significantly different response.
Line 227: Inhibited or reduced? / replace: that --> the Chl a/b ratio
Line 234 – 240: The two-way ANOVA also showed a significant interaction, meaning that both cultivars responded significantly different to Cd treatments for NTP production in root and shoot.
Line 241: Is it interesting to add a correlation analysis for both cultivars separately? Or is this a bad idea?
Line 248 – 253: Please inform the reader of Cd concentrations in the crops mentioned.
Line 259 – 260: Biomass reduction and TI give the same information, yet they are handled as different parameters.
Line 262: Bad syntax.
Line 267 – 268: Can you add shortly how Cd diminishes the capacity for ROS removal?
Line 273: You report oxidative damage. Please specify --> oxidative damage: oxidation of lipids
Line 273 – 274: How can a reduction in biomass be linked to oxidative stress?
Line 294: Specify which cultivar has constant or increased Cd TF.
Line 302 – 305: For me, the second part of the sentence is a harder to comprehend (…, whose tendency was the opposite of NPT content.). It might be just me, but maybe you want to have another look at it.
Line 306 – 308: Just thinking out loud: If high NPT in Idyll promotes Cd sequestration and cause it stay in the root, then you would expect Idyll to have higher Cd concentrations in the root compared to Harukaze. This is not the case! The higher NPT content might protect, but maybe Idyll also is an excluder which does not take up and transport Cd easily. This is just a suggestion and I am not stating this is correct, but you might want to look into some more literature.
Also, could it be that a lower translocation of Cd and higher amounts of NPT in the shoot are beneficial for the photosynthetic pigments? This might lead to a better photosynthesis, more energy and maybe better stress response?
By making this reasoning, I am assuming a lot of stuff and maybe the connections are not that straight or even true... It just thinking out loud. Maybe you want to dig into some more literature and also think about it. Please don't feel obligated.
Line 310 – 311: Unclear sentence.

Reviewer 2 ·

Basic reporting

The resubmitted manuscript is of excellent English quality, which is a major improvement in relation to the initial submission. In addition, the authors addressed the research question in a different way, not from a phytoremediation perspective but rather from a Cd tolerance/sensitive perspective and the mechanism behind this observation, which is much better when using a hydroponical setup for fundamental research.

Experimental design

OK

Validity of the findings

OK

Additional comments

The manuscript improved a lot with regard to the initial submission.

---

## Round 0.4 · Minor Revisions

Although all reviewers now basically endorsed your manuscript, there are a few new minor suggestions, from one of them. Therefore, I opted for minor revisions in order to allow you one last round of editing before it is Accepted and sent through to the editorial office.

Reviewer 1 ·

Basic reporting

Line 136-137: Are the wavelengths of 665, 649 and 470 respectively for chlorophyll a, b and carotenoids? Would be nice if this is clear for the reader by using "respectively" if it is.

Line 143: To what substances do these wavelengths correspond?

Line 141: "extraction" sounds like you are extracting MDA, where actually MDA is reacting with TBA (if I'm not mistaken...).

Line 143-145: "Lipid peroxidation was estimated with the content of MDA that is thiobarbituric acid-reacting substances (TBARS) as described by Ali et al. (2014)." --> I know what you mean, but the sentence is unclear. A suggestion: "MDA content was estimated through a TBARS assay, as described by Ali et al. (2014)."
+ I would consider making this sentence the first sentence of the paragraph.

Line 175: IC50 is underlined.

Line 280-281: "The oxidative lipids" sounds strange.

Line 309-311: "In our study, Cd concentration ..., the tendency of which was the opposite of NTP content...". This sentence is unclear to me. You might want to consider rephrasing.

Experimental design

No comment.

Validity of the findings

No comment.

Additional comments

In my opinion, using Cd contaminated biomass for biofuel might only be possible if filtering of the combustion gases takes place (or if Cd is not there in the biofuel after processing of the biomass). If not, isn't there a chance for rereleasing the Cd back into the environment during combustion of the Cd contaminated biofuel?

I enjoyed reviewing your article. Thank you for your effort in rewriting and considering my remarks.

---

## Round 0.5 · accepted · Accept

You have addressed all remaining minor comments. Therefore I'm happy to accept your manuscript for publication.